# Active RNAP pre-initiation sites are highly mutated by cytidine deaminases in yeast, with AID targeting small RNA genes

**Benjamin JM Taylor\*, Yee Ling Wu, Cristina Rada\***

Protein and Nucleic Acid Chemistry Division, Medical Research Council Laboratory of Molecular Biology, Cambridge, United Kingdom

**Abstract** Cytidine deaminases are single stranded DNA mutators diversifying antibodies and restricting viral infection. Improper access to the genome leads to translocations and mutations in B cells and contributes to the mutation landscape in cancer, such as kataegis. It remains unclear how deaminases access double stranded genomes and whether off-target mutations favor certain loci, although transcription and opportunistic access during DNA repair are thought to play a role. In yeast, AID and the catalytic domain of APOBEC3G preferentially mutate transcriptionally active genes within narrow regions, 110 base pairs in width, fixed at RNA polymerase initiation sites. Unlike APOBEC3G, AID shows enhanced mutational preference for small RNA genes (tRNAs, snoRNAs and snRNAs) suggesting a putative role for RNA in its recruitment. We uncover the high affinity of the deaminases for the single stranded DNA exposed by initiating RNA polymerases (a DNA configuration reproduced at stalled polymerases) without a requirement for specific cofactors.

## Introduction

Cytidine deaminases are a family of polynucleotide mutators that modify cytosines into uracil in viral nucleic acids as part of the innate immune defences (*Harris and Liddament, 2004*). Their success in restricting infection is reflected in the fact that the family has undergone a rapid expansion in primates and humans (*Jarmuz et al., 2002*). The ancestral founder of the family, activation induced deaminase (AID), functions in the adaptive immune system to mutate antibody genes in B cells as a fast mechanism to promote diversity of the antibody response to match the rapid evolution of pathogens during infection. The evolutionary advantages of these strategies are counterbalanced by the risk of exposing the host genome to active mutagenesis, a frequent cause of oncogenic transformation in leukaemia and lymphomas of B cell origin.

All members of the AID/APOBEC family are selective in the sequence context of the deaminated cytosine, with the two preceding nucleotides identifying the signature of individual deaminases (*Beale et al., 2004*). This mutation context signature has identified the human APOBEC3A and 3B proteins as the source of many of the somatic mutations accumulated by cancer genomes (*Nik-Zainal et al., 2012*; *Burns et al., 2013*; *Roberts et al., 2013*, *Taylor et al., 2013*). The combined mutational landscape observed in mammalian genomes is complicated by the contribution from multiple cellular processes in addition to enzymatic deamination, such as metabolic oxidation, methyl-CpG deamination and aging, thus elucidating the precise contribution of the APOBECs is far from straightforward (*Alexandrov et al., 2013*; *Lawrence et al., 2013*). However the peculiar clustering of same strand mutations at TpC dinucleotides observed in kataegic mutations in breast cancers constitutes a hallmark of the APOBEC3A and 3B deaminases that can be experimentally induced. Repair of double stranded DNA breaks can expose long patches of single stranded DNA with multiple deaminations leading to the mutation clusters observed in association with genomic rearrangements in breast cancer genomes (*Nik-Zainal et al., 2012*; *Roberts et al., 2012*; *Taylor et al., 2013*).

**\*For correspondence:** btaylor@
mrc-lmb.cam.ac.uk (BJMT); car@
mrc-lmb.cam.ac.uk (CR)

**Competing interests:** The authors declare that no competing interests exist.

**eLife digest** In cells, genetic information is stored within molecules of DNA, which contain sequences of four 'bases' arranged in different orders. Replacing one of these bases with a different base results in a mutation, which can have a positive or negative influence on the cell.

Mammals use a group of enzymes called cytidine deaminases to help defend themselves against harmful invaders. These enzymes work by introducing mutations into the DNA of viruses, microbes and even the mammal itself. For example, an enzyme called APOBEC3G can mutate the DNA of viruses to prevent them spreading around the body. Another enzyme, called AID, can mutate the genes that make antibodies—proteins that attack the invading microbes—in order to make new varieties of antibodies. Unfortunately, the enzymes sometimes target other genes, which can lead to cancer and other diseases.

Cytidine deaminases can only access and mutate single strands of DNA, so most of the DNA in a cell is protected because it is in a two-stranded double helix. However, there are times when the two strands are separated, such as when a section of DNA is being repaired, or when it is being transcribed to produce a molecule of RNA, which is subsequently used to make a protein. It is not clear when cytidine deaminases are able to target single stranded DNA, and whether they need help from any other components.

Now, Taylor et al. have studied how these enzymes access single stranded DNA when artificially introduced into yeast. These experiments showed that AID and APOBEC3G can access single stranded DNA without the help of any extra components. The enzymes target genes that are being transcribed to make RNA, with the DNA at the start of the transcription site being the most prone to mutation.

In mammal cells, most genes are normally protected from the mutations introduced by cytidine deaminases, but this protection does not appear to work in many cancer cells. The next challenge will be to develop a better understanding of how this protection works, and to work out why it sometimes goes wrong.

Physiologically, the activity of such mutators is targeted to specific substrates and restricted from the rest of the genome to limit genomic instability. In the case of AID, expressed upon activation in only a fraction of B cells, by limiting access to the nuclear compartment and preferential recruitment to the immunoglobulin genes; in the case of APOBEC3G, expressed preferentially in lymphoid cells, by its localisation in the cytosol and binding to the viral genome and capsid. The mechanism that preferentially directs AID to the immunoglobulin genes is not fully understood, but active transcription has been repeatedly invoked as a requirement (Reviewed in *Storb, 2014*) and many of the proteins found to be associated with AID are also involved in transcription and mRNA processing (*Pavri et al., 2010*; *Basu et al., 2011*; *Okazaki et al., 2011*; *Willmann et al., 2012*). Access of AID to off-target loci are documented not only by the anecdotal occurrence of mutations in oncogenes and chromosomal break points bearing the signature of the deaminase (Bcl6, MYC [*Pasqualucci et al., 2001*]) but also by AID dependent chromosome-break-capture and direct ChIP, where widespread off-target presence of AID is experimentally detected outside the immunoglobulin locus in mouse B cells (*Chiarle et al., 2011*; *Klein et al., 2011*; *Yamane et al., 2011*).

In addition to the sporadic off-target mutations induced by AID in B cells, APOBEC3A and 3B are thought to be responsible for many of the non-clustered/non-kataegic mutations at TpC dinucleotides observed not only in breast cancers but in other tumour types where the kataegic signature is not obviously present (*Kuong and Loeb, 2013*). As with sporadic AID mutations, the circumstances that promote or grant access of the APOBECs to single stranded DNA substrates of the host are not known. We have shown that overexpresion of deaminases in yeast faithfully recapitulates the mutation signatures observed in mammalian genomes. Here we have attempted to identify genomic features that promote or are permissive for enzymatic deamination by footprinting mutator activity on multiple genomes. Our results indeed reveal a preferential targeting of the deaminases to defined regions of the genome that is not dependent on cofactors but is rather based on accessibility, with structural features of the DNA at the promoter of actively transcribed genes being the key determinant. We also uncover a potential mechanistic explanation for the targeting and off-target preferences of the antibody diversification mutator AID.

# Results

## AID and APOBEC3G extensively mutate the yeast genome

Overexpression of cytidine deaminases in yeast leads to the accumulation of genome wide mutations, which can be monitored by the number of cells resistant to the arginine analogue L-Canavinine through inactivation of the arginine permease CAN1 gene (*Figure 1A*). We have previously shown that such over-expression leads to an uracil-DNA glycosylase (UNG) dependent enrichment of kataegic mutations through deamination of cytosines on single stranded DNA intermediates during the repair of double strand breaks (*Taylor et al., 2013*). To assess the distribution of isolated mutations, we obtained a dataset largely devoid of kataegic mutations by expressing the deaminases in *ung*Δ cells. Overexpression of AID* (an AID hyperactive mutant [*Wang et al., 2009*; *Taylor et al., 2013*]) in haploid cells results in highly elevated frequency of Canavinine resistant colonies ($164 \times 10^{-6}$), but relatively few mutations, averaging 61 single nucleotide variations (SNVs) per genome (*Figure 1A,B*). Diploid cells can overcome this limit as they avoid the reduction in fitness costs caused by accumulated mutation (*Waters and Parry, 1973*; *Lada et al., 2013*). Our experimental setting confirms this effect; whereas the mutation frequency is reduced almost 40-fold due to the requirement to inactivate both CAN1 alleles, the genome wide SNV increase over 10-fold, averaging 796 SNVs per genome for AID* and 592 SNVs for transformants expressing sA3G* (a hyperactive mutant of the catalytic domain of human APOBEC3G [*Wang et al., 2009*]; *Figure 1A,B*). For comparison, a database of mutations at C•G pairs was generated using the alkylating agent ethyl methane sulfonate (EMS). Alkylation of guanosines promotes base pairing with thymine, thereby causing G > A transitions during replication. Overnight exposure of diploid cells to 0.2% EMS resulted in increased mutation frequency and SNV load per genome similar to that elicited by the deaminases (*Figure 1*).

When interrogating the mutations (99.8% of which occur at C:G pairs; A:T mutations were excluded from further analysis; all detected mutations are given in *Supplementary file 1*), the expected flanking sequence context of WRC was found for AID* and YCC for sA3G* (*Figure 1C*). In stark contrast, no consensus motif was observed in the EMS data, highlighting the random nature of this mutagenesis. In all three datasets SNVs appeared distributed throughout the genome, with all chromosomes displaying similar overall mutation that is strongly correlated with chromosome length, ruling out major biases in the targeting of mutations (Spearman's correlation coefficient for AID*: $\rho > 0.65$; for sA3G*: $\rho > 0.55$; for EMS: $\rho > 0.68$; *Figure 1D*).

## Deaminase induced mutations are highly enriched in a small fraction of the genome

Whilst mutations are equally distributed amongst chromosomes, they are not uniformly arranged along the chromosome. By combining the SNVs from independent transformants, regions can be observed in AID* and sA3G* genomes which show pronounced mutational peaks (*Figure 2A*). Only one such region of high mutation density is seen in the EMS treated clones, that of the CAN1 gene. The presence of multiple loci with high mutation density is therefore a deaminase specific process.

A more detailed look at regions with high density of mutations reveals narrow peaks of accumulated mutation that are in many cases common to both deaminases (*Figure 2B*), with the most prominent peaks resulting from the proximity of several regions of densely targeted loci. These peaks represent high mutation densities within a bin size of 150 base pairs but surprisingly reflect the accumulation of mutations focussed to very narrow intervals within targeted loci (*Figure 2C,D*).

To further delineate mutation favoured loci, we defined regions of high mutation density by identifying overlapping 150 base pair fragments containing higher than expected mutation loads (minimum of six mutations per fragment, originating from three independent transformants). We identify 1227 and 568 such mutation-enriched loci (MELs) in the AID* and sA3G* treated genomes, in contrast to just 1 obtained for EMS treatment (overlapping the body of the CAN1 gene and hence due to canavinine selection). On average 35 such MELs would be expected for simulated datasets of equivalent mutation loads (*Figure 2E* and *Supplementary file 2*). MELs span remarkably narrow regions, with a window width averaging 110 bp for AID* and 71bp for sA3G* (*Figure 2F*), and with almost 41% of all AID* and 22% of all sA3G* induced mutations localised to these regions (*Table 1* and *Supplementary file 2*). In total, 25,618 of the combined 72,196 AID* and sA3G* mutations are occurring in MELs which account for just 1.5% of the genome (*Figure 2G*).

Both AID and APOBEC3G target cytosines for deamination within a specific sequence context, leading to the mutation hotspots associated with antibody diversification and the recurrent mutations

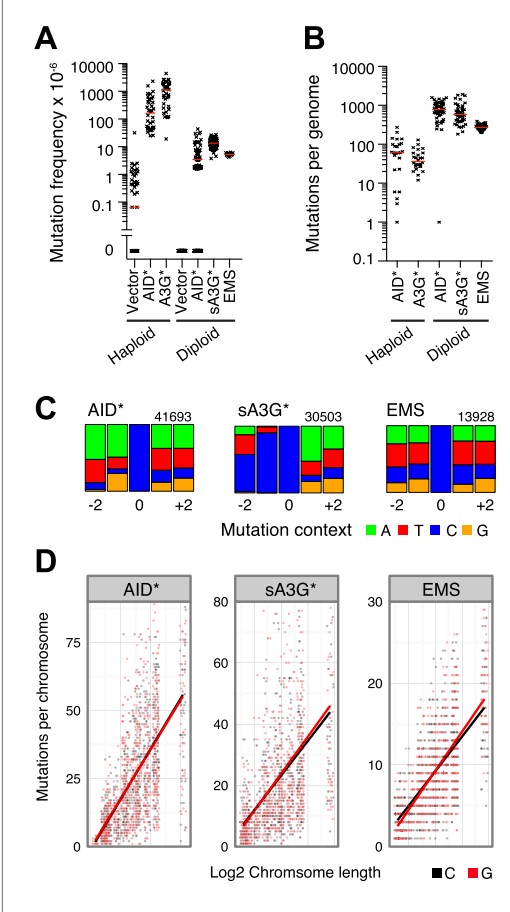

**Figure 1**. Genome wide distribution and signature of unclustered deaminase induced mutations in *ung1Δ* diploid yeast. (**A**) Mutation frequency (expressed as the number of canavinine resistant colonies per $10^6$) at the CAN1 locus in *ung1Δ* haploid yeast (data in part from *Taylor et al., 2013*) and *ung1Δ/ung1Δ* diploid yeast transformants expressing AID/APOBEC proteins or upon treatment with 0.2% EMS. Red bars indicate the median mutation frequency (n = 12–126 colonies). (**B**) Genome wide SNV number in *ung1Δ* haploid and *ung1Δ/Δ* diploid yeast transformants expressing AID/APOBEC proteins or with EMS treatment. Red bars indicate the median mutation per genome (n = 25–50 independent clones). (**C**) Sequence context of mutations at G•C pairs in diploid yeast genomes (indicated as mutations at cytosines) exposed to AID*, sA3G* or EMS mutagenesis. The numbers indicate total mutations per dataset, with the height of colour bars proportional to the frequency of each base found in the vicinity of a mutation. (**D**) Distribution of mutations per diploid yeast chromosome expressed as the number of mutations per chromosome in each independent genome against the chromosome length. The bars represent the projected linear trend for mutations at C (in black) or G (in red).

at CCC trinucleotides observed in HIV-1 genomes during the evolution of viral clades and which accumulate in viral genomes from infected individual (*Kijak et al., 2008*). We therefore analysed the distribution of AID and APOBEC3G preferred sequence context in the yeast genome and find that the densities of AID and APOBEC3G motifs (WRC and YCC respectively) show no enrichment within the highly targeted regions compared to the remaining genome (*Figure 2H*). Therefore, the accumulation of mutations in MELs is not a consequence of localised clustering of mutable motifs.

Reinforcing the notion that MELs are highly favoured targets for mutations, we find these areas are frequency mutated on both alleles: 48% of AID* genomes and 56% of sA3G* genomes have mutations within MELs occurring on both chromosome alleles, compared to just 2–3% predicted for random fragments of equivalent size and mutation loads. MELs also contain most of the homozygous mutations detected (82% of AID* and 78% of sA3G*). Targeting of both alleles in MELs suggests they represent highly mutable regions within the genome, with the deaminases returning repeatedly to the same sites (albeit on a second chromosome) to mutate.

Re-analysis of deaminase mutations we previously reported in haploid yeast (*Taylor et al., 2013*) identified 39 MELs which overlap with hypermutated MELs in diploid yeast, thus the focusing of mutations to MELs is seemingly unaffected by ploidy, suggesting the skewing of mutations due to selective pressures, such as fitness, is negligible (*Figure 2—figure supplement 1*). Equally, we observe no significant strand bias in the hypermutated hotspots associated with AID* MELs suggesting that both strands are targeted in a similar fashion. A broader distribution of sA3G* strand bias more likely reflects the partial skewing in the presence of YCC motifs at MELs (*Figure 2—figure supplement 2*).

In conclusion, deaminases preferentially target narrow focussed regions throughout the genome independent of the sequence density of deaminase targets.

## MELs exclusively overlap gene promoters

There is a well-recognised relationship between AID induced mutations and transcription both in B cells at immunoglobulin genes, and for off-target loci, with mutations preferentially accumulating towards the promoter proximal region of the transcription unit (*Pasqualucci et al., 2001*; *Rada and Milstein, 2001*). The transcription link is interpreted as a mechanism that facilitates access of AID due to the generation of single stranded DNA intermediates

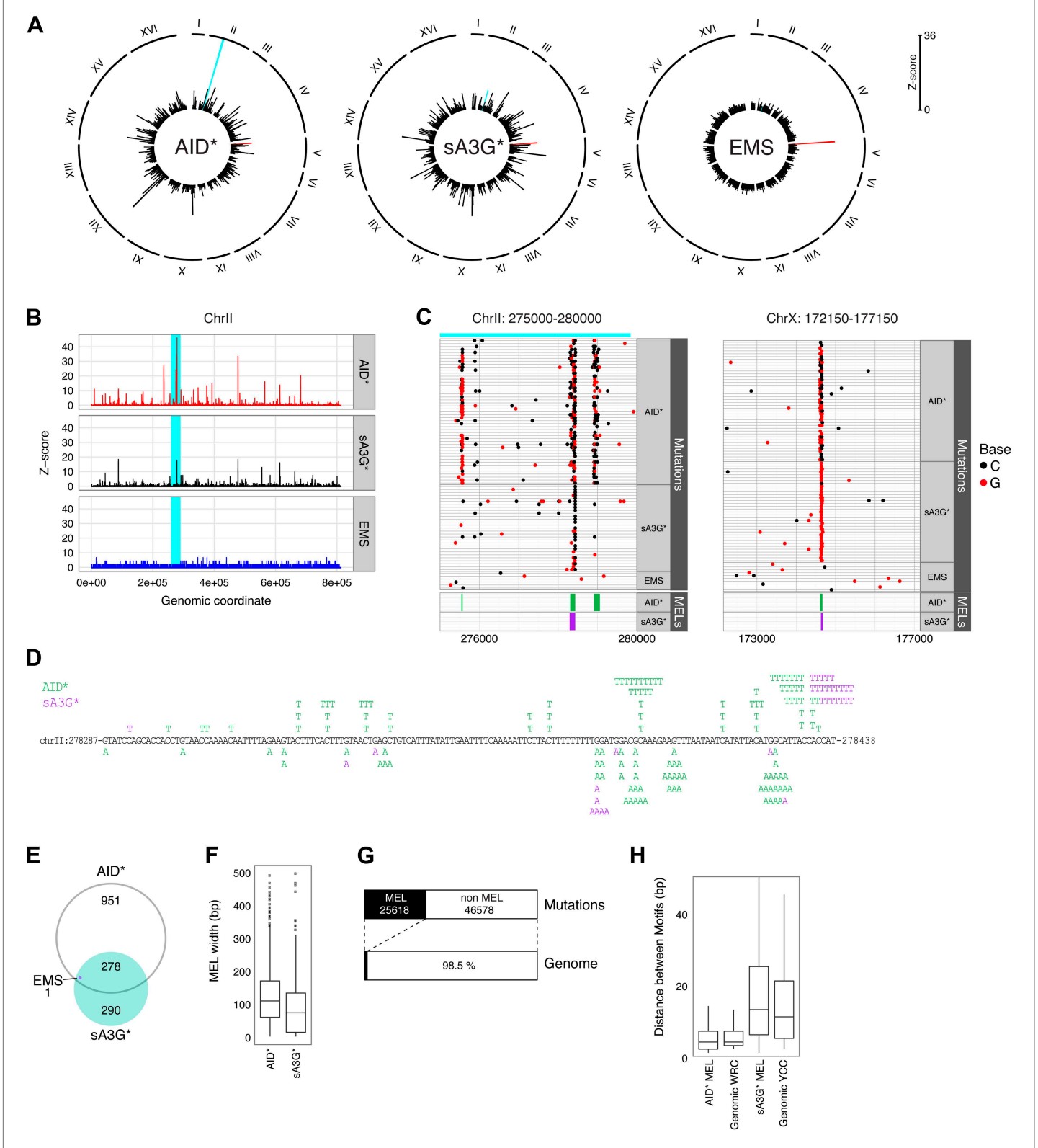

**Figure 2**. Mutation enriched loci (MELs) identified by focussed deaminase-induced mutation. (**A**) Radial histograms depict the density (Z-score) of pooled mutations for each dataset in 2 kb overlapping genomic segments along each chromosome. The CAN1 locus is highlighted in red. The peak highlighted in cyan is further enlarged in panels (**B**), (**C**) and (**D**). (**B**) Mutation densities along ChrII in AID* (red), sA3G* (black) and EMS (blue) treated

*Figure 2. Continued on next page*

*Figure 2. Continued*

genomes, expressed as the Z-score of mutation density per dataset (y-axis) along chromosome II (x-axis; 200 bp bin size). The region shadowed in cyan is magnified in (**C**). (**C**) Regions of high mutation density identify narrow mutation enriched regions (MELs), shown as green boxes for AID* and purple boxes for sA3G* in the bottom panel. Horizontal lines represent a single genome with each non-clonal mutation at C or G indicated by a dot (black or red respectively). Regions in Chr II and Chr X containing mutation enriched loci shown at the same scale, with the genomic coordinates indicated. (**D**) Mutations in the pronounced MEL on ChrII (highlighted cyan in panels (**A**), (**B**) and (**C**) shown in green for AID* and purple for sA3G*. Coordinates are indicated. (**E**) Overlap of detected MELs in AID*, sA3G* and EMS datasets. (**F**) Distribution of MELs width with the median indicated for AID* and sA3G* mutated genomes. (**G**) Fraction of the total deaminase mutations in MELs (black boxes) relative to genomic coverage of MELs. (**H**) Distribution of distances between AID and A3G mutable motifs within MELs vs genome wide mutable motif distances.

The following figure supplements are available for figure 2:

**Figure supplement 1**. Overlap between Haploid and Diploid MELs.

**Figure supplement 2**. Strand bias in deaminase induced mutations calculated as fraction of mutations at C (+strand) or G (- strand) within each MEL.

**Table 1.** Deaminase induced Mutation Enriched Loci (MEL) in yeast genomes

|  | Observed | | | Simulated | | |
|---|---|---|---|---|---|---|
|  | AID* | sA3G* | EMS | AID* | sA3G* | EMS |
| MELs | 1227 | 568 | 1 | 50 | 21 | 3 |
| % MEL mutation | 40.7 | 21.6 | 0.24 | 0.75 | 0.39 | 0.14 |

(**Chaudhuri et al., 2003**). We therefore wondered whether AID induced MELs would be found associated with transcription. Contrary to expectation, enrichment analysis reveals that both AID* and sA3G* MELs are depleted within the body of RNA polymerase II (RNAP II) transcribed mRNA genes and rather that the deaminase induced mutations are preferentially associated with promoter regions, with over 76% of deaminase targeted hotspots found at promoters, compared to just 24% for simulated fragments (**Figure 3A**).

Initiation of replication also transiently generates single stranded DNA at defined locations. However, there is no enrichment of mutated hotspots associated with replication origins (ARS) (**Figure 3A**). Although this could reflect the relative depletion of mutable motifs within ARS core consensus sequence, we find similar densities of mutable cytosines within the broader sequence context encompassing 200–300 base pairs nucleosome depleted regions associated with functional origins (**Figure 3—figure supplement 1**), suggesting that single strand availability provided by melting the DNA by the ORC complex might not be sufficient to efficiently target the deaminases.

Mutation enrichment at promoters is not restricted to hotspots identified within MELs, which exclude 73% of the total mutations due to the threshold applied in defining enriched loci. Aligning all mutations to mRNA transcriptional starts (TSS) and termination sites (TTS) (**Xu et al., 2009**), revealed a strong association of deaminase induced mutations with the TSS, with over 57% of AID* and 46% of sA3G* mutations occurring within the promoter region (defined as 500 base pairs upstream and 50 base pairs downstream of the TSS), compared to only 21% of EMS mutations (the expected frequency for randomly distributed mutations). Mutation accumulation is skewed upstream of the TSS (peak at -21 bp and -38 bp for AID* and sA3G* respectively; **Figure 3B**), corresponding to the nucleosome free region where the pre-initiation RNAP complex forms before scanning for the TSS (**Rhee and Pugh, 2012**). Indeed, aligning SNVs to the TATA box/TATA-like element or TSS revealed that not only are the majority of promoter associated mutations occurring between these two features (**Figure 3C**), there is also a paucity of mutations at the TATA-element suggesting this region is protected by TBP/TFIID binding (this paucity is, at least for AID*, not due to an absence of mutable sequence motifs; **Figure 3D**). Intriguingly, the peak of AID* and sA3G* induced SNVs centred 30 base pairs from the TATA-element, the region where TBP guides TFIIB to load RNAPII for the formation of the pre-initiation complex (PIC) (**Rhee and Pugh, 2012**).

The deaminase mutated hotspots thus identify the position where promoter melting occurs before the scanning polymerase encounters the TSS, suggesting a mechanistic basis for the hypothesis that the deaminases access the promoter coincidentally with the assembly of the pre-initiation complex. Consistent with the notion that initiating polymerases create transient access for the deaminases rather than specifically loading the proteins, we detect robust association of RNAP II with the promoter region of deaminase targeted promoters in yeast but negligible enhancement in the association of

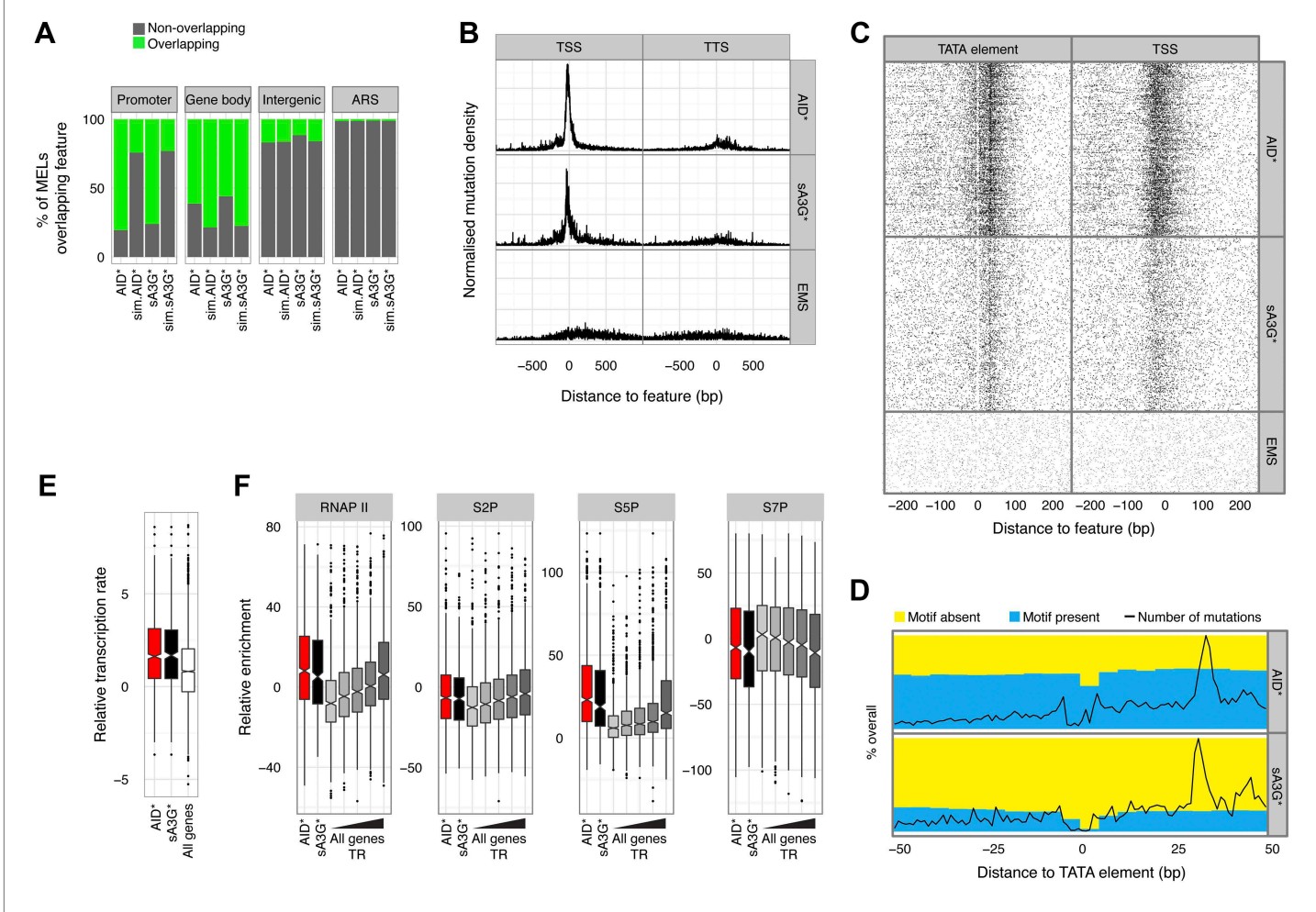

Figure 3. Deaminase mutation footprints are focussed to the pre-initiation complex region of active promoters. (A) Proportion of promoters, gene bodies, intergenic regions and replication origins (ARS) harbouring a MEL (green) or not (grey) for AID* and sA3G* datasets vs the expected distribution (sim.AID*sA3G*) determined by Monte Carlo simulation of equivalent sized fragments for each MEL dataset distributed randomly across the genome. (B) Density of mutations in relation to their distance to the nearest transcription start site (TSS) of mRNA (RNAP II) transcripts compared to the density relative to transcription termination sites (TTS). Data includes all mutations in addition to MELs. (C) Deaminase mutations relative to the TATA or TATA-like element for each RNAP II promoters (*Rhee and Pugh, 2012*) compared to the mutation distance distribution aligned to the transcription start site (TSS). (D) Proportion of AID* or sA3G* mutable motifs within RNAP II promoter regions, centred on the TATA-elements (*Rhee and Pugh, 2012*). Total number of mutations for each dataset is shown at each position (black line). (E) Relative transcription rates (see methods) at RNAP II promoters targeted by MELs compared to relative transcription rates for all RNAP II genes in gal induced conditions (*García-Martínez et al., 2004*). (F) Relative enrichment of RNAP II and RNAP II CTD phosphorylation (S2P, S5P and S7P) in promoters containing AID* (red) and sA3G* (black) MELs and all RNAP II promoters (grey) ranked according to transcriptional activity (*García-Martínez et al., 2004*).

The following figure supplements are available for figure 3:

**Figure supplement 1**. Paucity of deaminase mutations at replication origins is not a consequence of absence of mutable motifs.

**Figure supplement 2**. Density of mutations in relation to their distance to the nearest TATA box or TATA-like element.

**Figure supplement 3**. Distribution of the deaminases on chromatin is unrelated to mutation preferences.

**Figure supplement 4**. Transcription factor binding sites compared to MEL preferences.

either AID or sA3G with mutated promoters compared to unmutated or intergenic regions (*Figure 3—figure supplement 3*). Additionally, while there is a correlation between the mutated strand and the direction of transcription (*Figure 2—figure supplement 2*), MELs are predominantly composed of mutations occurring in both strands suggesting the PIC makes both strands available during initiation.

Supporting the idea that the deaminases preferentially mutate promoters due to their ability to recognize the melted DNA associated with the transcription pre-initiation complex, we observe that MELs occur in genes with above average transcriptional activity (*García-Martínez et al., 2004*) but targeting appears unrelated to any particular transcriptional program (*Figure 3—figure supplement 4*). Rather than simply transcription factor binding at the promoter, active initiation by RNAP II is important for MEL development (Wilcox test p < 0.005 for all groups; *Figure 3E*). The transition of RNAP II from the pre-initiation complex to the elongation complex is associated with a shift in phosphorylation of the C-terminal domain (CTD) serine 5/7 to serine 2 (*Kim et al., 2010*). In agreement with the transcription rate analysis, deaminase MELs are associated with both high levels of RNAP II occupancy and CTD-S5P, that parallels the association with the highest transcribed genes (*Figure 3F*). Indeed, the recurrent association of both AID* and sA3G* MELs with regions enriched for the basal transcription machinery and in particular Spt16 -a chromatin chaperon associated with highly transcribed genes (*Formosa, 2013*) (*Figure 3—figure supplement 4*), reinforces the idea that active transcription and potential pausing (at promoters highly dependent on the FACT/Spt16 complex) determines the deaminases targeting.

In summary, cytidine deaminases mutate at specific loci through the yeast genome, predominantly within active gene promoter regions.

## AID targets promoter regions of small RNAP III genes

In B cells, AID is found in association with components of the transcription machinery such as SPT5 and SPT6, and RNAP II itself (*Nambu et al., 2003*; *Pavri et al., 2010*; *Okazaki et al., 2011*), therefore we wondered whether the enrichment of mutations associated with promoters might be a feature restricted to RNAP II dependent genes. Analysis of mutations in highly transcribed non-RNAP II dependent transcripts, such as RNAP III dependent tRNA genes, astonishingly reveals an even more pronounced enrichment of targeted hotspots with 78% of the genomic regions corresponding to tRNAs harbouring repeated mutations. While we find that both sA3G* and AID* MELs overlap with tRNAs, AID* MELs are disproportionately overrepresented, with 228 of 275 tRNA genes being highly targeted (*Figure 4A*). Furthermore, aligning of mutations within 250 base pairs of the TSS of tRNA genes shows that all occur within the tRNA gene body, which is also the site of RNAP III initiation (*Figure 4B*). As in the case of mRNA promoters, the mutations in tRNAs are highly focussed to narrow hotspots that span the site where loading of the polymerase is thought to occur (*Figure 4C*).

The mutation frequency (normalised number of mutations per 550 base pairs) in AID* genomes within tRNA genes is higher than at mRNA gene promoters (p value < 2e-16, Wilcox non-parametric test; *Figure 4C*) and much higher than that observed even in the subset of highly transcribed mRNA promoters. While the differences in mutation frequency between mRNA promoters and tRNAs is still statistically significant for A3G* (p value < 8e-10), this effect is less pronounced. Enhanced mutation is also observed in the promoters of snoRNA and snRNA genes, again particularly in the case of AID* genomes, whereas no statistically significant differences are observed between any of the promoter subsets for mutations driven by EMS. The enhanced mutation attributable to AID* is not likely a feature of RNAP III, since snoRNAs are even more targeted for mutation though all but snR52 are transcribed by RNAP II (*Moqtaderi and Struhl, 2004*).

Targeting of tRNA, snRNA and snoRNA genes by the deaminases could be enhanced by the availability of hypermutable motifs, as there is on average one more YCC motif in the tRNA genes (1.5 more in the MEL region itself) targeted by sA3G* than in those tRNA genes not targeted by sA3G*. We see no such difference with AID* target motifs which are present within tRNA, snRNA and snoRNA gene promoters at similar frequency as in other promoters (average 52 to 63 motifs per 550 base pairs promoter window, *Figure 4—figure supplement 1*). Overall, there is only weak correlation between the number of motifs within the 550 base pair promoter window and the number of mutations (Spearman's ρ = 0.02 and ρ = 0.2 for AID* and sA3G* respectively; *Figure 4—figure supplement 2*), confirming that motif availability is not the main determinant for targeting.

Mutations at rRNA genes were poorly mapped due to the repetitive nature of the region on Chr XII (150–200 copies of the 9.1 kb unit containing the 35S pre-RNA and the 5S RNA). By including repeatedly mapped reads across the rDNA locus, we could detect several hundred mutations at low allele

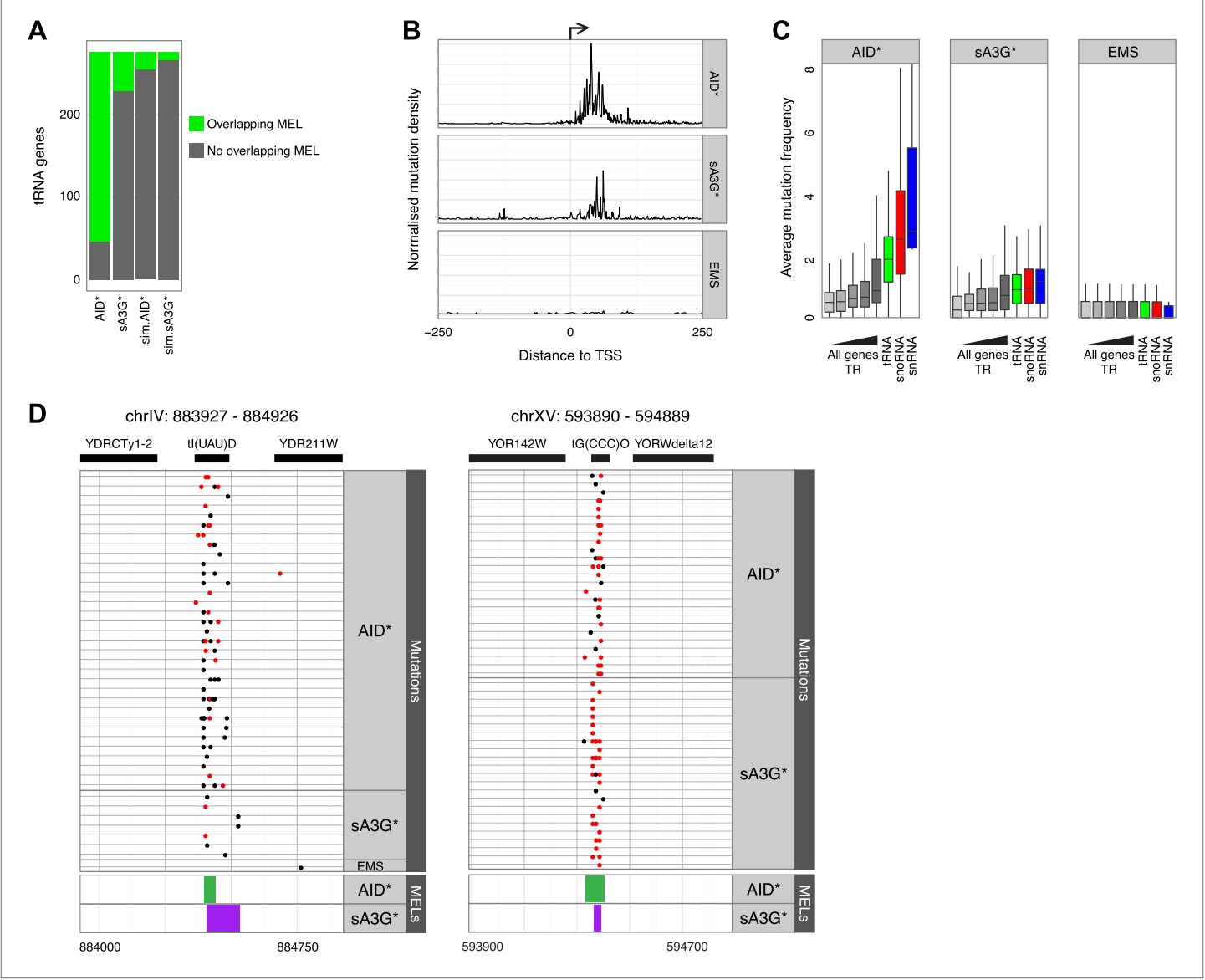

**Figure 4**. AID* and sA3G* target both RNAP II and RNAP III promoters. (**A**) Number of tRNA genes harbouring (green) an AID* or sA3G* MEL compared with expected number from Monte Carlo simulations. (**B**) Density of mutations in relation to the transcription start site (TSS) of tRNA genes. Mutations within the 500 base pair interval centred at the TSS are included. (**C**) Mutation frequency in promoters of mRNA genes (within a window 500 bp upstream and 50 bp downstream of the TSS) compared to the frequency of mutations in the promoters of tRNA (550 bp window centred on the middle of the tRNA gene), snoRNAs and snRNA genes (550 bp window as for mRNA genes). mRNA genes are binned according to transcription rate as in *Figure 3*. Both RNAP II and III driven snoRNAs are included. (**D**) Example of MELs in ChrIV and ChrXV corresponding to tRNA tI(UAU)D and tG(CCC)O, depicted as in *Figure 3*.

The following figure supplements are available for figure 4:

**Figure supplement 1**. Median number of mutable motifs in promoter regions.

**Figure supplement 2**. Mutationally enriched loci are not a consequence of increased density of mutable motifs.

**Figure supplement 3**. Mutations in the rDNA locus are restricted to the replication fork block (RFB) site.

**Figure supplement 4**. Deaminase induced mutation distribution in relation to R-loop forming potential.

frequency all within the expected deaminase mutation context, giving confidence in their detection and location (*Figure 4—figure supplement 3A*). Mutations were restricted to the well defined ribosomal replication fork barrier (rRFB) located between the 5S and 35S transcriptional units. No enhanced mutation was detected at the promoter regions (which are transcribed in opposite directions by RNAP III and RNAP I respectively). However mutations clustered at the rRFB site for both deaminases (*Figure 4—figure supplement 3B*), at a site where induced homologous recombination maintains the size of the ribosomal gene array. Although DNA double-strand breaks (DSB) have been detected at the site, it is likely that in vivo persistent breaks are rare in undamaged yeast (*Fritsch et al., 2010*). Accordingly we did not detect kataegic like clusters in the region, but rather localised mutated hotspots. Thus it is possible that other mechanisms such as cryptic transcription (*Houseley et al., 2007*) might expose the site to the action of the deaminases, rather than repair of double strand breaks. While AID overexpression in yeast deficient for components of the RNA processing machinery (THO) have enhanced genomic instability, particularly in highly transcribed GC-rich regions prone to R-loop formation (*Gómez-González and Aguilera, 2007*), in wild type yeast this effect is only mild. Nonetheless we observe positive association of MELs with predicted R-loop potential genes although the paucity of these features across the genomes (between 59–78 sites) precludes any predictive dissociation between high density of mutation, R-loop potential and transcription rates (*Figure 4—figure supplement 4*).

## AID but not sA3G binds small RNAs

An alternative explanation for the enhanced targeting of small RNA promoters by AID* is that the RNAs themselves preferentially bind AID, thereby creating co-transcriptional enrichment of AID in the vicinity of their genes. Purified AID binds RNA, with its in vitro deamination activity enhanced by treatment with RNAse A (*Bransteitter et al., 2003*), whereas the non-catalytic domain of APOBEC3G is responsible for its ability to bind RNA and form high molecular weight ribonucleic–protein complexes (*Huthoff et al., 2009*; *Bélanger et al., 2013*). It is not known whether binding in both cases is specific for any particular RNA species, but based on our current observations we decided to test the ability of human AID and human APOBEC3G to bind in vitro transcribed tRNA as well as polyU RNA. Whereas both Flag-tagged overexpressed human AID and full length human APOBEC3G can be recovered from cell extracts by binding to biotin labelled RNAs, the catalytic domain of APOBEC3G (sA3G) is not (*Figure 5A*). Furthermore, full length APOBEC3G is efficiently recovered from extracts by the extended linear polyU RNA, a reflection of its ability to oligomerise in an RNA dependent fashion, whereas AID recovery is not enhanced by its binding to linear polyU RNA. Binding of AID to tRNA species was also found for endogenous yeast tRNAs, suggesting that the modifications found in vivo (pseudouridylation and 2'-O-ribose methylation) do not affect the interaction. The single domain APOBEC3A protein shows no RNA binding ability except a limited amount to doubled stranded RNA, despite sharing the preferential targeting to promoters as the rest of the deaminases (*Figure 5—figure supplement 1A*). Taken together, this data suggest a degree of specificity in the RNA binding preferences of the deaminases, with AID preference linked to structured rather than linear RNA (*Figure 5B*). Interestingly the catalytic activity of AID is not required for the binding or the specificity, since similar binding was observed for the inactive mutant AID-E58A (*Figure 5A*).

In order to test the RNA binding properties of AID in modulating its targeting preferences we introduced a chimeric snR6 RNA into the RNAP II driven YBR194W gene, which was identified in our dataset as a transcribed but poorly targeted promoter by both deaminases (*Figure 5C* top panels). Initiation and transcription of the modified locus remained overall unaffected (*Figure 5—figure supplement 2*), while comparison of the YBR194W promoter region by Sanger sequencing revealed enhanced mutation focused to the immediate vicinity of the TSS by AID* but not sA3G*. No such focussing of mutations was observed in the unmodified yeast overexpresing AID* (*Figure 5C*).

We conclude that the differential preference of AID* for tRNA, snRNAs and snoRNAs in yeast might reflect the ability of AID to preferentially associate with abundant small RNA species, in contrast to the catalytic domain of APOBEC3G (sA3G*) that possesses no RNA binding activity.

Targeting mutations to initiating promoters is not likely a function of the size of the deaminase, as could be inferred from the results described for both AID and the single domain fragment of APOBEC3G used in our study. Similar promoter associated recurrent mutations can be elicited not only by APOBEC3A (also a single domain deaminase) but also by the double domain APOBEC3B (*Figure 5—figure supplement 1A*). It is therefore not entirely unexpected to observe enrichment of mutations at TpC (versus other dinucleotides) in association with promoter regions in a breast cancer genome that has the

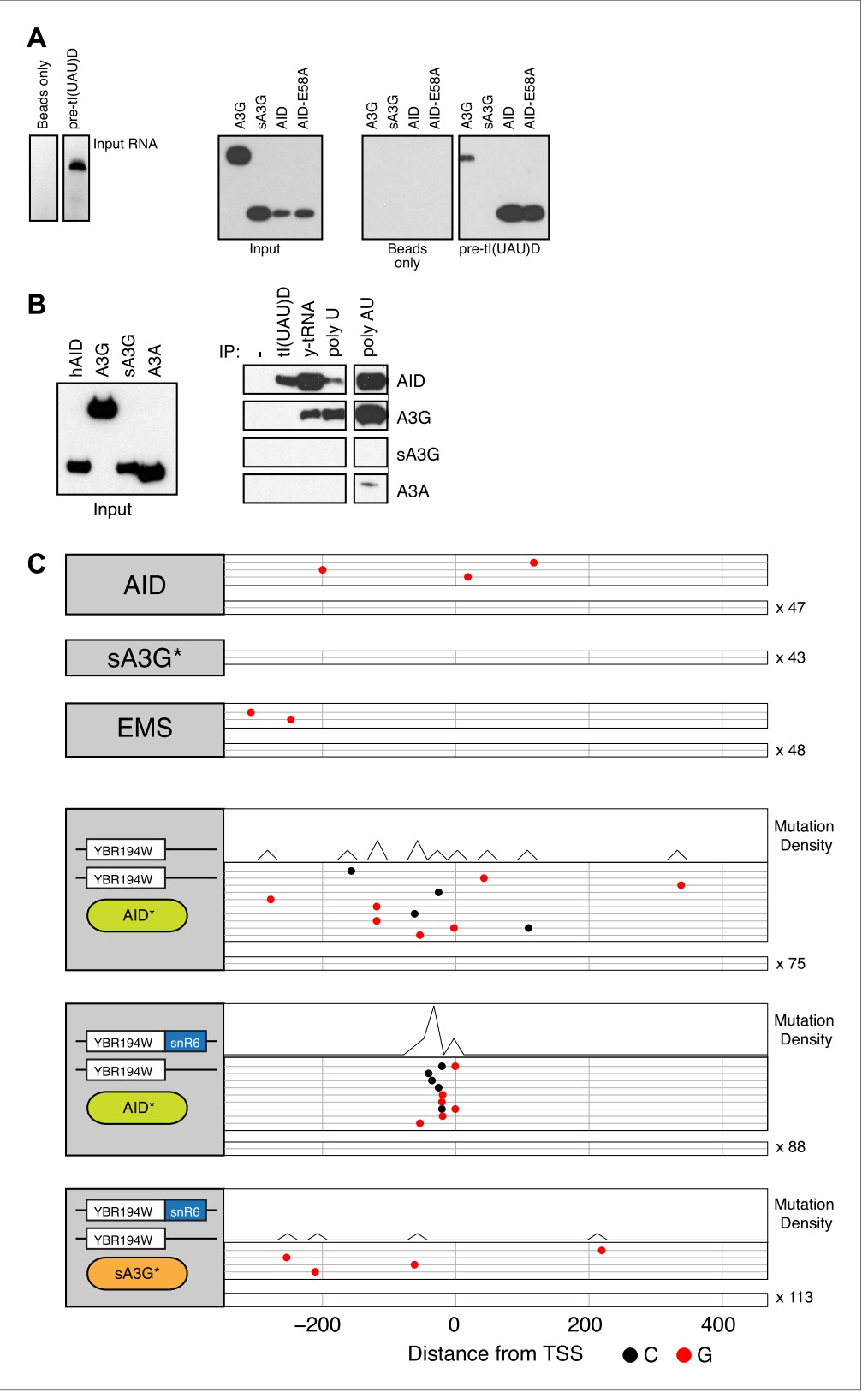

**Figure 5**. RNA binding by human AID and APOBEC3G. (**A**) Left panel shows the in vitro transcribed pre-tI(UAU)D tRNA used for affinity purification. Right panel shows immunoblots for transiently overexpressed AID/APOBEC3G
*Figure 5. Continued on next page*

*Figure 5. Continued*

proteins following RNA-immunoprecipitation with pre-tRNA. (**B**) Affinity purification with tl(UAU)D probe, total yeast tRNA, homopolymeric single stranded (polyU) and double stranded (polyA:U) RNA. Left panel shows input proteins, right panel shows immunoblots for transiently overexpressed AID/APOBEC3 proteins following RNA-immunoprecipitation. Results representative of at least 3 independent experiments. (**C**) Deaminase induced mutations in the promoter region of the YBR194W locus. Top panels: accumulated mutations in the AID*, sA3G* and EMS whole genome datasets. Bottom panels: mutations detected in Sanger sequenced yeast clones unmodified or harbouring a chimeric YBR194W-snR6 locus. Each line represents one clone with dots representing mutations (at C, black; at G, red). Clones with no mutations are indicated.

The following figure supplements are available for figure 5:

**Figure supplement 1**. Promoter mutations are driven by APOBEC3A and 3B and are a feature of cancer genomes enriched for TC mutations.

**Figure supplement 2**. Functional comparison of the YBR194W locus in modified yeast clones.

---

highest incidence of APOBEC3 kataegic mutations (*Figure 5—figure supplement 1B*), suggesting that the deaminases could access dsDNA at initiating or paused RNAPs also in mammalian cells.

## Discussion

The involvement of AID and APOBEC3A and 3B in cancer suggests that enzymatic deamination of genomic targets is an infrequent but recurrent consequence of the presence of the deaminases in vertebrates. Despite subcellular compartmentalisation, specific targeting and restricted expression limiting AID off-target activity, some genomic regions other than the natural target, the immunoglobulin loci, are predisposed to mutation. BCL6, PIM1 and MYC are recurrent off-targets of AID mutation in B cell malignancies (*Pasqualucci et al., 2001*); in the case of BCL6 it is estimated that AID induced mutations are also prevalent in non transformed B cells at just $10^3$-fold lower frequency than at immunoglobulin genes (*Liu et al., 2008*) and even in the absence of the mutator phenotypes attributable to malignant transformation, normal B cells frequently show AID induced translocations at the MYC locus (*Roschke et al., 1997*; *Casellas et al., 2009*). In cancer genomes, the association of APOBEC mutations with genomic rearrangements suggests that replication stress, persistent DNA lesions and incomplete repair expose single stranded DNA that becomes a substrate for deaminases leading to clustered mutations. It is unclear how APOBEC3A and 3B gain access to single stranded DNA leading to the singlet isolated mutations highly prevalent in mutated cancer genomes that bear the APOBEC signature (*Taylor et al., 2013*). It is therefore important to understand the genomic context that facilitates off-target activity of the deaminases in the absence of explicit DNA damage.

Expression of AID and other APOBEC proteins in yeast faithfully recapitulates the signature of mutations observed in mammalian cells in a smaller genome with no background mutations due to unrelated processes, such as DNA repair (*Lada et al., 2013*; *Taylor et al., 2013*). In this study we demonstrate the non-random nature of the mutations induced by the deaminases, which is remarkably focussed to just 1.5% of the yeast genome but nonetheless overlaps more than half of the active promoters. AID is known to interact with components of the transcription machinery in mammalian cells (reviewed in *Kenter, 2012*). However, the overlap between highly mutated promoters by both AID and APOBEC3G suggests that rather than conservation of protein–protein interactions of the deaminases with the transcription complex, properties of the promoter itself can determine targeting.

Enhanced targeting of RNAP III transcribed genes argues against active recruitment of the deaminases by conserved initiation factors, whereas the structural conservation of the DNA template conformation at the core pre-initiation complex of all polymerases (*Vannini and Cramer, 2012*) supports the idea that the conformation of the DNA template is the common element in the recruitment of the deaminases. Indeed, the site of polymerase loading (within the body of the tRNA genes) rather than the TSS is the preferred target of deamination in the case of the RNAP III transcribed tRNAs in contrast to the 5′ region of the RNAP III transcribed *SNR52* snoRNA, where the loading of the RNAP is fixed at the 5′ promoter region. Furthermore, the high density of mutations focussed to the small region between the TATA binding protein site (TBP) and the transcription start site (TSS), more precisely identify the pre-initiation complex (PIC) as the target for the deaminases.

Budding yeast RNAP II promoters show characteristic and highly regulated nucleosome exclusion. This is partly due to sequence composition, with regions enriched for poly dA•dT nucleotides that confer rigidity to the DNA and are therefore thermodynamically less favourable to wrap around nucleosomes (*Yuan et al., 2005*), and partly due to the regulated and precise positioning of the +1 nucleosome relative to the TSS that includes specific histone variants (H2A.Z and H3.3) that promote chromatin accessibility (reviewed in *Jiang and Pugh, 2009*). Therefore it is highly significant that other nucleosome free regions, such as ARS are not targeted by the deaminases, despite undergoing DNA melting during the initiation of replication. This reinforces our interpretation that intrinsic properties of active promoters, in particular the configuration associated with loading of the polymerase at the pre-initiation complex (open pre-initiation complex) (*Grünberg et al., 2012*), are sufficient to generate persistent single stranded DNA accessible for deamination. Our data supports the presence of such open PICs in most yeast active promoters.

Neither the preferential targeting of promoters nor the narrow focus of the MELs is due to the preferential clustering of mutable motifs. Interestingly, the nature of the mutation hotspots within MELs (both at C and G), reveals that both strands of the melted DNA structure associated with active promoters are accessible. Furthermore, protection from mutation is evident at the TBP binding site while the peak of mutations ~30 base pairs downstream identifies the site of RNAP loading and DNA melting mapped by permanganate footprinting (*Giardina and Lis, 1993*) and high resolution ChIP (*Rhee and Pugh, 2012*). Our deaminase footprinting data further confirms the persistent open configuration and single stranded nature of this region potentially identifying open pre-initiation promoters.

Differences in the assembly of the PIC in TATA and TATA-like promoters, do not seem to affect mutation susceptibility, although predictably, TATA box promoters show a more defined distance between the TBP protected footprint and the accessible melted DNA (*Figure 3—figure supplement 2*) indicating that it is the structure of the single stranded DNA rather than the assembly (SAGA or THIID dependent) of the transcription initiation complex itself that determines targeting (*Rhee and Pugh, 2012*).

Up to 75% of human promoters in different cell types are occupied by a pre-initiating form of RNAP II (*Guenther et al., 2007*), whereas pausing and stalling are much more common in metazoan transcription compared with *Saccharomyces cerevisiae*. Mammalian promoters are frequently regulated by proximal pausing, with most promoters pausing within 200 base pairs of the TSS (*Adelman and Lis, 2012*). In the presence of a deaminase, initiating and or paused sites would become accessible for mutation, thus it is intriguing to observe promoter proximal enrichment of mutations at TpC dinucleotides in PD4120a, a breast cancer genome with dramatic accumulation of kataegis that betrays its mutagenesis by APOBEC3B (*Nik-Zainal et al., 2012*). Our data favours the idea that accessibility of single stranded DNA at RNAP II stalled sites suffice to recruit APOBECs or indeed AID. This model offers explanation for the association of AID with mammalian SPT5, which functions in modulating the pausing of RNAP II during elongation as transcription stalls, and is consistent with the recurrent targeting by AID of the promoter proximal region of MYC (*Duquette et al., 2005*) a well characterised promoter-proximal pausing regulated gene (*Krumm et al., 1992*; *Strobl and Eick, 1992*).

The correlation between high transcription rates and enhanced deaminase targeting reinforces the hypothesis that repeated loading of the pre-initiation complex leads to the persistence of a small region of melted DNA that is very efficiently targeted by the deaminases. Indeed the enhanced targeting of tRNA, snoRNAs and snRNA genes could reflect the high transcription rates of these essential RNAs given that RNAP I and III transcripts constitute almost 80% of the total nuclear gene expression in dividing cells (*Vannini, 2013*). The unexpected finding that tRNAs are disproportionally targeted for mutation by AID compared with APOBEC3G, as are the promoters of other highly structured RNAs (snRNA or snoRNA), and the indication that this difference is not due to motif enrichment at those promoters, brings into focus the potential involvement of the RNA binding properties of the deaminases in promoting targeting. While APOBEC3G has been shown to bind not only HIV RNA, but cellular RNAs, including abundant 7S RNA (*Huthoff et al., 2009*), this ability is dependent on the N-terminal domain. Mutation targeting of the RNAP initiation complex is not linked to the ability of the deaminases to bind RNA per se, as the catalytic C-terminal domain of APOBEC3G in this study is inert regarding RNA binding. Notably, our results show that AID binds structured RNAs in vitro (such as tRNAs), and preferentially targets tRNAs and other small RNA promoters for mutation in yeast, prompting the speculation that binding to abundant RNAs sequesters AID to subnuclear localities such as nucleolar areas, where small RNAs genes also localise during transcription. Indeed nucleolar localisation of overexpressed AID has been reported in mammalian cells, although its significance

under physiological levels remains to be tested (*Hu et al., 2013*). Alternatively preferential recognition of particular RNA structures such as folded tRNAs could determine the recruitment of AID to genomic regions.

In conclusion, our study uncovers the remarkable preference of mammalian cytidine deaminases to mutate active promoters when expressed in yeast, a preference blind to the type of RNA polymerase (both RNAP II and III genes are targets) and not ascribable to sequence context or targeting by specific cofactors. The precise and narrow location of the recurrent mutations pinpoints the site where the RNAP pre-initiation complex is loaded highlighting the conservation of the TBP (TATA binding protein) site and the formation of the pre-initiation complex, whereas exclusion of mutations from the TBP site confirms the poised nature of active yeast promoters.

These results suggest that initiating polymerases create a small but persistent accessible patch of single stranded DNA in vivo, which has high affinity for deaminases and where both strands are accessible for mutation. They also strongly support the notion that AID might directly bind to single stranded DNA at the pre-initiating or stalling RNAP sites without a requirement for specific cofactors and that its targeting is modulated by its ability to interact with structured RNA species.

## Materials and methods

### Yeast transformants

Yeast strain BY4743 *ungΔ/ungΔ* was generated by crossing BY4741 *ungΔ* (MATa; *his3Δ1*; *leu2Δ0*; *met15Δ0*; *ura3Δ0*) obtained from Euroscarf deletion collection (Frankfurt, Germany) with the BY4742 *ungΔ* strain. BY4742 *ungΔ* was generated by removal of the *UNG1* open reading frame by homologous recombination in the parental BY4742 strain, using a PCR generated *URA3* cassette flanked by a 57-bp 5′ homology and 51-bp 3′ homology arms that include adaptamers for post integration removal of the *URA3* selection cassette (*Reid et al., 2002*). The YBR194W-snR6 chimeric strain was generated by inserting a URA3 cassette at the 5′ end of the YBR194W gene in BY4741 *ungΔ* cells. Homology arms and the snR6 gene were amplified from genomic DNA using the primers (1) 5′-CCTGCCAC TTTCAAAAGGCG-3′ and 5′-CGAAGGGTTACTTCGCGAACTCCTGTCCCTATTACATATTCAACC-3′, (2) 5′-GGTTGAATATGTAATAGGGACAGGAGTTCGCGAAGTAACCCTTCG-3′ and 5′-GCCAGGCATGC TAATGGCAAAACGAAATAAATCTCTTTGTAAAAC-3′, (3) 5′-GTTTTACAAAGAGATTTATTTCGTTTTG CCATTAGCATGCCTGGC-3′ and 5′-TGGTGGTCATATGCTCGGTG-3′. A PCR fusion of all three fragments with the first and last primer was used to retarget the URA3 containing locus. 5-Fluoroorotic acid counter-selection was used to isolate targeted colonies that were then mated with BY4742 *ungΔ* to generate the final BY4743 *ungΔ/ungΔ* YBR194W-snR6/YBR194W strain. Correct integration of all targeting constructs was confirmed by PCR.

Yeast transformation and selection, genomic DNA extraction and mutation frequency calculation were performed as described previously (*Taylor et al., 2013*). Control and AID* expression vectors were as described previously (*Taylor et al., 2013*). The sA3G* vector was generated by PCR amplification of the C-terminal domain of A3G* fused with a 5′ SV40 nuclear localisation sequence and FLAG tag using primers 5′-GCAAGCTTGCCACCATGCCTAAAAAGAAGCGTAAAGTCGAGATTCT CAGACACTCG-3′ and 5′-CCAGAATCAGGAAAACGGAGCAGACTACAAGGACGATGACGACAAGTA GCTCGAGGC-3′ and ligating the resultant Hind III-Xho I fragment it into pRS426-GAL1pr-tADHpolyA vector described previously (*Taylor et al., 2013*).

Ethyl methanesulfonate (EMS) mutagenesis was performed by culturing BY4743 *ungΔ/ungΔ* yeast overnight in YEPD with 0.2% EMS, after which cells were washed in 5% sodium thiosulfate and plated for viability and canavanine resistance as above.

### Sample preparation and DNA sequencing

DNA libraries were generated using the multiplexing Nextera DNA Sample Prep Kit (Illumina, Little Chesterford, UK) according to manufactures instructions. The libraries were sequenced by BGI (BGI, Beijing, China). The de-multiplexed sequence reads were aligned to the reference yeast genome (SacCer_Apr2011/sacCer3) using BWA-MEM (*Li and Durbin, 2009*). Optical duplicates were removed using Picard (http://picard.sourceforge.net) and only uniquely mapped paired reads were retained. On average 43-fold sequence coverage was achieved for each yeast genome. Unprocessed sequence reads for this study have been deposited at the EMBL-EBI European Nucleotide Archive, study accession number PRJEB7456 (http://www.ebi.ac.uk/ena/data/view/PRJEB7456).

## Data analysis

### Mutation calling

An in-house pipeline for mutation calling was used where GATK base quality score recalibration and indel realignment (*McKenna et al., 2010*) was performed prior to somatic mutation calling by Somatic Sniper (*Larson et al., 2012*) using the parental BY4743 genome as reference. High confidence single nucleotide variations (SNVs) were filtered using the following criteria: (1) SomaticSniper score >50, (2) allele frequency ≥0.3, (3) reference or samples read count ≥4, (4) average position as fraction on reads ≥0.1, (5) average distance to 3′ end ≥0.1, (6) average base quality ≥30, (7) average read length >50 bp.

### Mutation enriched loci (MEL) identification

Within each data set, mutations were pooled with the number of mutations within 150 base pair windows. Based on the assumption of a random distribution of mutation amongst the fragments, a binomial distribution was determined using the following parameters: size equal to the average number of mutations per clone and probability equal to the average number of mutations per clone over the total number of mutable motifs. Mutable motifs were the total number of WRC, YCC, or C bases for AID*, sA3G* and EMS respectively. The 99th percentile was used as a threshold to identify significantly mutated windows and adjacent windows merged. To refine the span of each individual mutation enriched loci (MEL), unmutated residues and residues falling in the following categories were removed and the window size adjusted: bases that had a count below the 25th percentile of all the counts in the window; bases which had a mutation count below four standard deviations from the average for the window and all bases with only a single detected mutation (where the median mutation count was above one). A final threshold was applied so that only regions with more than 5 mutations derived from at least four independent transformants were assigned as high confidence MELs. All MELs were manually assessed using a genome browser and are shown in *Supplementary file 3*.

The averaged fraction of overlapping regions for simulated MEL dataset were determined by 1000 cycles of bootstrap analysis using randomised equivalent number of fragments of identical sizes for each dataset distributed across the genome.

### Normalised mutation density

The normalised mutation density was calculated by dividing the mutation count for each residue by the total number of mutation for the dataset.

### RNAP enrichment

ChIP enrichment was determined by taking the sum of the ChIP enrichment scores (*Kim et al., 2010*) for each promoter fragment (defined as 550 bp upstream and 50 bp downstream from the TSS [*Rhee and Pugh, 2012*]). Promoters were then grouped according to the transcription rate (*García-Martínez et al., 2004*) or whether they contained a MEL.

### Average mutation frequency for mRNA, tRNA, snoRNA and snRNA promoters

Promoter fragments for mRNA genes and transcription rate binning were performed as above. tRNA gene promoter fragments were defined as a 550 bp fragment centred on the middle of the tRNA gene. snoRNA and snRNA promoters were defined as 550 bp upstream and 50 bp downstream from the TSS defined in the *Saccharomyces* Genome Database (*Cherry et al., 2012*). Intronic snoRNA genes were assigned the mRNA promoter and polycistronic snoRNA genes were assigned only one promoter. Mutation frequency was calculated by first randomly down-sampling the databases to half the size of the EMS dataset, to allow equivalent numbers of mutations to be compared. The number of mutations occurring on each promoter was then calculated. The process was bootstrapped 1000 times to give a directly comparable average number of mutations for each promoter.

### rDNA mapping

To detect mutations at the repetitive rDNA locus a less stringent algorithm was used. De-multiplexed sequence reads were aligned as before and unmapped reads removed. Reads mapping to the rDNA region (chrXII:434839-508289) were extracted and used for mutation calling by SomaticSniper. Mutations with a SomaticSniper score of above 50, a read depth of 10 in both the reference and the sample and no evidence of the mutated base in the reference genome were assigned.

All analyses were performed using Bioconductor. Scripts are included as *Supplementary file 4*.

## Immunopreciptation

### RNA binding

The tl(UAU)D RNA probes were generated by in vitro transcription (MegaShortScript T7 Kit, Life Technologies, Paisley, UK) with or without biotin-UTP (Life Technologies), according to manufactures instructions. Free nucleotides were removed using Oligo Clean & Concentrator columns (Zymo, Irvine, CA, USA). The tl(UAU)D template was generated by annealing the following oligos 5'-AATTTA ATACGACTCACTATAGGGCTCGTGTAGCTCAGTGGTTAGAGCTTCGTGCTTATAACG-3' and 5'-TGCT CGAGGTGGGGTTTGAACCCACGACGGTCGCGTTATAAGCACGAAGCTCTAACC-3'. The pre-tl(UAU) D template was generated by PCR amplification from yeast genomic DNA using the following primers 5'- AATTTAATACGACTCACTATAGGGCTCGTGTAGCTCAGTGGTTAGAGC-3' and 5'-TGCTCG AGGTGGGGTTTGAACCCACGACGG-3'. Biotinylation of total yeast RNA (Life Technologies), polyu-ridylic acid, polyadenylic acid-polyuridylic acid (Sigma–Aldrich, Gillingham, UK) and the tl(UAU)D probe were performed using the RNA 3' End Biotinylation Kit (Pierce) according to manufacturers instructions.

Biotinylated RNA probes (3.6 μg) were refolded by heating to 80°C for 5 min in folding buffer (25 mM Tris pH 7.6, 100 mM KCl, 1 mM EDTA), $MgCl_2$ was then added to a final concentration of 20 mM and the RNA allowed to slowly cool to 10°C before being bound to magnetic beads (Pierce, Loughborough, UK) for 1 hr at 4°C. Unbound probe was removed by washing with RNA buffer (25 mM Tris pH 7.6, 50 mM KCl, 5 mM NaCl, 1.5 mM $MgCl_2$, 35 mM Glycine, 10% glycerol) supple-mented with 0.5% Triton X-100. The integrity of the RNA was monitored by denaturing gel electro-phoresis and staining with toluidine blue.

Clarified whole cell extracts (in RNA buffer supplemented with 0.3% Triton X-100 and complete protease inhibitors [Roche, Burgess Hill, UK]) from HEK 293 cells expressing Flag-AID, catalytically inactive AID (E58A mutation), APOBEC3G-Flag and the SV40-NLS tagged catalytic C-terminal domain of APOBEC3G (sA3G)-Flag, were incubated for 1 hr at 4°C in the presence of bead bound biotinylated RNA probes. Unbound proteins were removed by washing the beads four times in RNA buffer sup-plemented with 0.5% Triton X-100 at 4°C and the bound protein monitored by western using anti Flag antibodies (M2-HRP, Sigma–Aldrich).

### Chromatin imunoprecipitation

Overnight 60 ml yeast cultures fixed in 1% formaldehyde for 20 min and quenched in 0.125 M glycine (final) were washed twice in cold PBS, resuspended in RIPAlo (150 mM NaCl, 10 mM Tris–HCl pH 7.5, 1 mM EDTA, 1% Triton X-100, 0.1% SDS, 0.1% Sodium Deoxycholate, 1× Complete protease inhibi-tors) prior to lysis using a MPI TissueLyser (10 cycles of 30 s on, 5 min off, 4000 rpm), sonication using a Bioruptor (14 cycles of 30 s on, 30 s off, high intensity) and centrifugation (10 min 15,000×g). Equal amounts of clarified chromatin were incubated overnight at 4°C with 3 μg anti-HA 16B12 (Covance, Maidenhead, UK), 2 μg anti-H3 ab1791 (Abcam, Cambridge, UK), 2 μg anti-RNAPII S5P ab5131 (Abcam). Purification followed on Protein-G dynabeads for 2 hr with extensive washes (twice in RIPAlo, twice in RIPAhi [RIPAlo but for 500 mM NaCl], once in RIPA-LiCl [RIPAlo but 250 mM LiCl replacing NaCl] and twice in TE) and overnight elution in 25 mM Tris–HCl, 1 mM EDTA, pH to 9.8, 50 μg/ml proteinase K at 65°C. Input DNA was extracted using Gentra Puregene (Qiagen, Manchester, UK) with qPCR performed using QuantiFast SYBR kit (Qiagen) all as per manufactures instructions. Primers used are; YBR019C; 5'-ATCCAGCACCACCTGTAACC-3' and 5'-AAACTTCTTTGCGTCCATCC-3', YBR020W; 5'-ACCTGAGTTCAATTCTAGCGC-3' and 5'-TCCGGTTTAGCATCATAAGCG-3', YNL067W; 5'-AACCAAACTCTAGCCTCCAA-3' and 5'-TGCTGACAGTAACACCTTCTGG-3', YBL003C; 5'-TGTGC ACTCTACCAACTGGG-3' and 5'-ATGTCCGGTGGTAAAGGTGG-3', YPL250C; 5'-AGAGAGTTGCTCC AGACCCT-3' and 5'-GCATAAAGAAGCGGCTCTGC-3', YEL009C; 5'-GGGGGAGAGTAACCTGTGTT-3' and 5'-TTTCGGCTCGCTGTCTTACC-3', YBR194W; 5'-TCTTCTTGCTCGGGGTTCTC-3' and 5'-TGCTG AAGGCCTTTGCAAAG-3', YPL189W; 5'-GCGAAGATTACGGCACTCGA-3' and 5'-ACAGGTACGGGC TATCTGGA-3', YLR183C; 5'-ACATCTGCCACGACACATCA-3' and 5'-TGGTGGAGAGTACGGATCCA-3', YJL105W; 5'-TTTCTTGCTCTTGGCGGCTA-3' and 5'-AGTTAGGATCTGAGCCGGGT-3', YPR007C; 5'-ACAGGTTCGAGCTTCATGGG-3' and 5'-CGGGAATTTCATCCAGCGGA-3', chrXV;367475-367594 5'-ACTTGGCACTTCTTCCTCAACA-3' and 5'-TCGCAAAGTTGGCTAACCGT-3', chrX;585916-586020 5'-ATGTCTCCCTGTTACCCGGT-3' and 5'-ACAGGTGCTGTCACAAAACA-3', chrIV;76,875-76,955 5''-GGCAGCACCGAGAATGTTTT-3' and 5'-GCTGTTAGCATATTGGGGGT-3'.

## Yeast transcript analysis

RNA from 1 ml overnight cultures purified with RNAeasy plus (Qiagen) was used to generate cDNA using oligo-dTs and the GoScript Kit (Promega, Southampton, UK) followed by qPCR employing QuantiFast SYBR (Qiagen) all as per manufactures instructions. Primers used are TAF10; 5′-ATATTCCAGGATCAGGTCTTCCGTAGC-3′ and 5′-GTAGTCTTCTCATTCTGTTGATGTTGTTGTTG-3′, ACT1; 5′-CTTTCAACGTTCCAGCCTTC-3′ and 5′-CCAGCGTAAATTGGAACGAC-3′, YBR194W-snR6; 5′-CCTGCCACTTTCAAAAGGCG-3′ and 5′-CAGGGGAACTGCTGATCATCTCTG-3′, YBR194W; 5′-GGGTCGTGAAAAAGAGAACGG-3′ and 5′-ATGTGATGGTGCAGTGCCTC-3′.

## YBR194W promoter sequencing

The YBR194W promoter region was amplified using the following primers; 5′-ATTGTGGCAGTTC GGCTTTG-3′ and 5′-AGGTTTCCCAGTCTGGCTTG-3′ and Sanger sequenced using the latter.

## Acknowledgements

We are grateful to David Rueda and Myron Goodman for sharing unpublished results and members of the Rada lab for helpful advice and discussions. The late Michael Neuberger instigated the initial stages of this work and remains in memory an inspiration. This work was supported by the Medical Research Council (MRC reference number MC_U105178806) and through an MRC Centennial Award to BJMT.

## Additional information

### Funding

| Funder | Grant reference number | Author |
| --- | --- | --- |
| Medical Research Council | MC_U105178806 | Benjamin JM Taylor, Yee Ling Wu, Cristina Rada |
| Medical Research Council | Centenary Award | Benjamin JM Taylor |

The funder had no role in study design, data collection and interpretation, or the decision to submit the work for publication.

### Author contributions

BJMT, Conception and design, Acquisition of data, Analysis and interpretation of data, Drafting or revising the article; YLW, Acquisition of data, Analysis and interpretation of data, Drafting or revising the article; CR, Conception and design, Analysis and interpretation of data, Drafting or revising the article

### Author ORCIDs

Benjamin JM Taylor, http://orcid.org/0000-0001-6101-3786
Cristina Rada, http://orcid.org/0000-0003-4898-5550

## Additional files

### Supplementary files

• Supplementary file 1. Catalogue of yeast mutations.

• Supplementary file 2. Coordinates of MELs.

• Supplementary file 3. All mutationally enriched regions (MELs). Top panel indicate position of each non-clonal mutation indicated by a dot (at C, black; at G, red), with horizontal lines representing a single genome. Middle panel shows MELs (AID*, green; sA3G*, purple; EMS, grey). Bottom panel displays genomic features (including transcripts, replication origins, centromers), coloured according to feature type, with arrows indicating the direction of transcription. The coordinates of the region are indicated. Regions are ranked according to the number of mutations present.

• Supplementary file 4. Scripts used for data analyses.

## Major datasets

The following dataset was generated:

| Author(s) | Year | Dataset title | Dataset ID and/or URL | Database, license, and accessibility information |
|---|---|---|---|---|
| Taylor BJM, Wu YL, Rada C | 2014 | Data from: Active RNAP pre-initiation sites are highly mutated by cytidine deaminases in yeast, with AID targeting small RNA genes | PRJEB7456; http://www.ebi.ac.uk/ena/data/view/PRJEB7456 | Publicly available at EMBL-EBI European Nucleotide Archive (http://www.ebi.ac.uk). |

The following previously published datasets were used:

| Author(s) | Year | Dataset title | Dataset ID and/or URL | Database, license, and accessibility information |
|---|---|---|---|---|
| Ludmil B Alexandrov, Serena Nik-Zainal, David C Wedge, Samuel A Aparicio, Sam Behjati, Andrew V Biankin, Graham R Bignell, Niccolò Bolli, Ake Borg, Anne-Lise Børresen-Dale, Sandrine Boyault, Birgit Burkhardt, Adam P Butler, Carlos Caldas, Helen R Davies, Christine Desmedt, Roland Eils, Jórunn Erla Eyfjörd, John A Foekens, Mel Greaves, Fumie Hosoda, Barbara Hutter, Tomislav Ilicic, Sandrine Imbeaud, Marcin Imielinski, Natalie Jäger, David T Jones, David Jones, Stian Knappskog, Marcel Kool, Sunil R Lakhani, Carlos López-Otín, Sancha Martin, Nikhil C Munshi, Hiromi Nakamura, Paul A Northcott, Marina Pajic, Elli Papaemmanuil, Angelo Paradiso, John V Pearson, Xose S Puente, Keiran Raine, Manasa Ramakrishna, Andrea L Richardson, Julia Richter, Philip Rosenstiel, Matthias Schlesner, Ton N Schumacher, Paul N Span, Jon W Teague, Yasushi Totoki, Andrew N Tutt, Rafael Valdés-Mas, Marit M van Buuren, Laura van 't Veer, Anne Vincent-Salomon, Nicola Waddell, Lucy R Yates, Australian Pancreatic Cancer Genome Initiative,  ICGC Breast Cancer Consortium, ICGC MMML-Seq Consortium, ICGC PedBrain, Jessica Zucman-Rossi, P Andrew Futreal, Ultan McDermott, Peter Lichter, Matthew Meyerson, Sean M Grimmond, Reiner Siebert, Elías Campo, Tatsuhiro Shibata, Stefan M Pfister, Peter J Campbell and Michael R Stratton | 2013 | Data from: ftp://ftp.sanger.ac.uk/pub/cancer/AlexandrovEtAl | 10.1038/nature12477 | Publicly available. |
| García-Martínez J, Aranda A, Pérez-Ortín JE | 2004 | Data from: http://scsie.uv.es/chipsdna/chipsdna-e.html#datos | 10.1016/j.molcel.2004.06.004 | Publicly available. |
| Kim H, Erickson B, Luo W, Seward D, Graber JH, Pollock DD, Megee PC, Bentley DL | 2010 | Data from: http://downloads.yeastgenome.org/published_datasets/Kim_2010_PMID_20835241/ | 10.1038/nsmb.1913 | Publicly available. |

| Rhee HS, Pugh BF | 2012 | Data from: http://downloads.yeastgenome.org/published_datasets/Rhee_2012_PMID_22258509/ | 10.1038/nature10799 | Publicly available. |
|---|---|---|---|---|
| Xu Z, Wei W, Gagneur J, Perocchi F, Clauder-Münster S, Camblong J, Guffanti E, Stutz F, Huber W, Steinmetz LM | 2009 | Data from: http://downloads.yeastgenome.org/published_datasets/Xu_2009_PMID_19169243/ | 10.1038/nature07728 | Publicly available. |
| Venters BJ, Wachi S, Mavrich TN, Andersen BE, Jena P, Sinnamon AJ, Jain P, Rolleri NS, Jiang C, Hemeryck-Walsh C, Pugh BF | 2011 | Data from: http://downloads.yeastgenome.org/published_datasets/Venters_2011_PMID_21329885/ | 10.1016/j.molcel.2011.01.015 | Publicly available. |

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
