## [Decision Letter]

Thank you for sending your work entitled “Active RNAP pre-initiation sites are highly mutated by cytidine deaminases in yeast, with AID targeting small RNAs genes” for consideration at *eLife.* Your article has been favorably evaluated by James Manley (Senior editor) and 2 reviewers, one of whom, Nick Proudfoot, is a member of our Board of Reviewing Editors.

The Reviewing editor and another expert reviewer discussed their comments before we reached this decision, and the Reviewing editor has assembled the following comments to help you prepare a revised submission.

Analysis of the in vivo effect of human deaminases, such as AID and APOBEC3G, in heterologous systems such as *E. coli* and yeast has been undertaken by different labs and has contributed greatly to understand their mechanisms of action. Despite the limitations that yeast may have as a system to make conclusions about the in vivo action of these deaminases in human cells, it is clear that this type of approach reveals important features of the genome accessibility for these enzymes known to act specifically on ssDNA. In yeast, the pattern of mutations induced by these deaminases has been reported by different laboratories. This manuscript of Taylor and Rada, from the former laboratory of M. Neuberger, extends such studies further. A detailed analysis of clustering of mutations along the genome of diploid yeast reveals that such clusters are preferentially located at promoters of highly expressed RNAPII-driven genes. This is the case for both AID and APOBEC3G-induced mutations, and in the case of AID this is also observed for RNAPIII genes. Taylor and Rada conclude that this is due to the ability of the small RNAs to bring AID to its site of action. To probe this hypothesis they show that AID binds in vitro to yeast tRNA and poly U. This work is interesting and novel by providing new information about the yeast genome accessibility to human deaminases that should help clarify its mechanism of action. It is consistent with the ability of these enzymes to act on ssDNA formed at active pre-initiation complexes.

1) We are not convinced that the authors provide significant evidence that this recruitment is RNA mediated. AID was discovered by the Honjo lab as an RNA modifying enzyme, and the ability of AID to interact in vitro with RNA is not that novel. More importantly, the data presented is insufficient to prove that RNA plays a direct role in recruiting AID to pre-initiation complexes in vivo. The difference between RNAPII and RNAPIII promoters (embedded in the body of the genes) could be due to a different structure of the pre-initiation complex, a difference in the opening and accessibility of the ssDNA formed at such promoters or to different factors that might influence the action of AID in vivo, not necessarily its recruitment. Despite this, it is clear that these proteins act at RNAPII promoters as deduced from the impressive enrichment of MELs at such promoters. Is the average length of the RNA generated in such promoters small? Do long RNA molecules bind AID in vitro?

2) It is important to distinguish loading and recruitment of the deaminases from their sites of action (as seen by their mutational analysis). This study can only deduce sites of action not sites of loading and recruitment. This later point should be addressed by performing ChIP analysis to show that the deaminases are loaded at the promoter by the promoter initiation complex. Also reporter systems could be employed to probe these issues in vivo by turning on or off transcription (using inducible promoters e.g. GAL) This is relevant given the reported role of transcription factors in AID recruitment in human cells that in principle would not correlate with mutational hotspots.

3) Are the authors sure that Canavinine is an Arginine permease transporter inhibitor or just an Arginine analogue that uses the same permease (encoded by the CAN1 gene). This later reason could explain why can1 mutations impede canavarine uptake leading to can resistance? Please comment.

4) Can the authors further explain why mutations within the MELs are much more likely to occur on both alleles of the diploid strain compared to equivalent random fragments? As the information on whether such mutations are coincident between both alleles is not clear, I wonder whether the authors can exclude the possibility that such high levels could be due to gene conversion type of events whether or not mediated by double-strand breaks.

5) It is important that 57% of AID and 46 % of sA3G mutations occur within the promoter region compared to only 21% of EMS mutations. However, considering that the authors found 1227 and 568 MELs in the AID and sA3G treated genomes, but only 1 for MMS, a simple analysis would reveal that the MMS mutations in the promoter region compared to the total frequency of MMS-induced MELs is extremely high. Could the authors clarify or discuss this point better, so that the reader does not get misled by a simple analysis, if that is the case?

---

## [Author Response]

We have reinforced our conclusion that in yeast, the main determinant for targeting of the deaminases is the accessibility of the single stranded substrate at preinitiation complex sites. We now include ChIP data on the chromatin association of AID and APOBEC3G (sA3G) and new bioinformatic analyses. We have also reinforced the evidence in support for a role of RNA binding by AID in modulating the preferred sites of mutation by the addition of new genetic data, ChIP data and expression analysis. We have made changes in the text to improve the clarity and incorporate the new data, which has also resulted in the addition of a new author.

*1) We are not convinced that the authors provide significant evidence that this recruitment is RNA mediated. AID was discovered by the Honjo lab as an RNA modifying enzyme, and the ability of AID to interact in vitro with RNA is not that novel. More importantly, the data presented is insufficient to prove that RNA plays a direct role in recruiting AID to pre-initiation complexes in vivo. The difference between RNAPII and RNAPIII promoters (embedded in the body of the genes) could be due to a different structure of the pre-initiation complex, a difference in the opening and accessibility of the ssDNA formed at such promoters or to different factors that might influence the action of AID in vivo, not necessarily its recruitment. Despite this, it is clear that these proteins act at RNAPII promoters as deduced from the impressive enrichment of MELs at such promoters*.

Both sA3G and AID (small ∼27Kd proteins) can access and mutate DNA at promoters; however we noted that AID showed marked preference for promoters of small structured RNAs (such as tRNAs or snRNAs). The two proteins are a perfect genetic control for each other, since the most salient difference between the two deaminases is their ability to bind RNA: the catalytic domain of APOBEC3G (sA3G) does not bind RNA, while AID seems to have a very good ability to bind RNA and in particular small structured RNAs such as tRNAs or snRNAs. We speculated that this RNA binding difference must be the main determinant for the different targeting preferences of AID.

In order to confirm this finding, we have now transplanted a small structured RNA to a locus that although transcribed was not a preferred target for either AID or sA3G. The results are included in a revised Figure 5, and show enhanced targeting of the promoter of the modified loci by AID but not sA3G. A similar number of new sequences confirm that the unmodified promoter remains a low frequency/untargeted locus for AID. Although at this moment we do not fully understand the mechanism for this recruitment, we think it confirms our general inference that AID targeting *in vivo* is affected by it ability to interact with RNA.

*2) It is important to distinguish loading and recruitment of the deaminases from their sites of action (as seen by their mutational analysis). This study can only deduce sites of action not sites of loading and recruitment. This later point should be addressed by performing ChIP analysis to show that the deaminases are loaded at the promoter by the promoter initiation complex*.

We agree with the main concern of the reviewers in that we believe the activity of the deaminases does not necessarily correlate with their presence at chromatin, however we feel it is unlikely that ChIP experiments will provide a definitive answer to dissecting the mechanism that brings the deaminase to a genomic region (loading) versus the mechanism that promotes active mutation. The current literature in mammalian cells regarding association of AID with chromatin (as measured by ChIP) is complicated. The most solid data shows association of AID at the Sµ switch repeat region [a region that is highly enriched for AID targets] which is regulated by phosphorylation (Vuong et al 2013). ChIP data only occasionally and haphazardly correlates the presence of AID with mutation at off-target sites ([62], Hogenbirk et al 2012, discussed in Vaidyanathan et al 2014), unless other factors such as persistent RNAP II stalling are also present.

We have performed chromatin immunoprecipitation for both AID and the sA3G proteins. Our results are now included in Figure 3—figure supplement 3, where they confirm the expected absence of specific association of the deaminases with the promoters. The signal for the deaminases is found at similar frequency within both mutated and unmutated promoters and at unmutated intergenic regions.

We interpret our ChIP data as evidence for the transient nature of the interaction between the deaminase and its substrate in yeast [probably reflecting a high k-off], unlike mammalian cells where additional interactions might stabilize the presence of the deaminases (such as specific targeting factors at immunoglobulin genes or interaction with mammalian Spt5 or RPA30).

We have argued in the past that mutation is a more reliable measure of the “past” association of the deaminases with their substrate. At this moment we do not have a reliable assay to measure unproductive association of the deaminases that would distinguish occupancy of a locus versus functional outcome. Single molecule analysis might be a possible way to compare recruitment and mutation, but we feel those experiments are beyond the scope of this manuscript.

We are careful in our manuscript to avoid making the claim that the promoter initiation complex “actively loads” the deaminases, since we rather interpret our findings as indicating that the conformation of the DNA at the promoter site is permissive for the “access and activity” of the deaminases.

Also reporter systems could be employed to probe these issues in vivo by turning on or off transcription (using inducible promoters e.g. GAL) This is relevant given the reported role of transcription factors in AID recruitment in human cells that in principle would not correlate with mutational hotspots.

We have analysed transcription factor binding data and find enrichment of members of the basal transcription machinery and associated chromatin remodelling factors such as Spt16. No specific transcription factor is particularly associated with MEL containing promoter regions. The results are included as a new Figure 3—figure supplement 4, highlighting the association of deaminase mutation with highly transcribed genes. We believe these analyses support the main conclusion that the structure of the substrate at transcription initiation sites is the main determinant for off-target mutation by the deaminases, rather than particular association with a family of transcription factors.

Is the average length of the RNA generated in such promoters small? Do long RNA molecules bind AID in vitro?

As indicated in the modified Figure 5, both the full double domain version of APOBEC3G and AID bind poly U RNA (0.3 – 2 kilobase pairs), although APOBEC3G is consistently better in our assays at binding “long” RNA whereas AID is better at binding short structured RNAs like the tRNAs. All deaminases tested are able to bind double stranded polyA:U RNA, including the single domain deaminase APOBEC3A.

We have repeated the immunoprecipitation experiments testing the RNA binding properties of the deaminases to make the result as robust and reproducible as possible and have modified Figure 5 with updated blots that reflect at least 3 independent experiments.

Our data suggests that different deaminases vary in their preferences and capacity to bind single, double stranded or highly structured RNA. Our results agree with evidence from the literature that suggests APOBEC3G might multimerise on diverse messenger RNAs. Further studies will focus on the preferences of AID-RNA interactions in mammalian cells.

As to whether there is a gene length preference for deaminases, for AID* and sA3G* datasets, we find no particular bias for mutations to occur at short or long RNAP II mRNA genes (see Figure 6 below).Author response image 1.

3) Are the authors sure that Canavinine is an Arginine permease transporter inhibitor or just an Arginine analogue that uses the same permease (encoded by the CAN1 gene). This later reason could explain why can1 mutations impede canavarine uptake leading to can resistance? Please comment.

We apologise for the mistake in our description of mutation the CAN1 locus as a selection system and thank the reviewer for pointing it out. As the reviewers rightly suggest, we use resistance to L-canavanine (a toxic arginine analogue) that is imported inside the cell by the yeast *S. cerevisiae* using the amino acid transporter encoded by the CAN1 gene. Mutations that impair the transport also render the cells resistant to the toxicity of the drug. We have now corrected the text.

4) Can the authors further explain why mutations within the MELs are much more likely to occur on both alleles of the diploid strain compared to equivalent random fragments? As the information on whether such mutations are coincident between both alleles is not clear, I wonder whether the authors can exclude the possibility that such high levels could be due to gene conversion type of events whether or not mediated by double-strand breaks.

We observe that in many cases the same DNA fragment is associated with high density of mutations (MELs) in both alleles of the same gene (frequently but not always at the same nucleotide position). We argue the simplest explanation is repeated targeted of both alleles, either simultaneously or in successive rounds of mutation. Regions of equivalent size as MELs would only be expected to show biallelic mutations (homozygous or heterozygous) in 2-3% of the fragments (as predicted from a random distribution of the same number of mutations as in the observed datasets). We have modified the text in an attempt to make the point more clear but keeping the text concise.

As the reviewers mention, an alternative that we do not favour is that gene-conversion between the two alleles accounts for the high frequency of homozygous mutations in MELs. We repeatedly discuss the absence of kataegic mutations as an indication that DNA breaks and repair are very rarely found in association with the mutation preferences in our datasets. Our published data (53) demonstrated that DNA breaks associated with deamination are frequent in Ung+ wild type cells and results in clustered kataegic mutations but rare or absent in Ung- yeast. Given that repair by gene conversion would rely on a break of the DNA, the lack of kataegic mutations argues against a mechanism that would require breaks in association with MELs.

Since we favour repeated targeting by the deaminases as the explanation for the high frequency of bi-allelic mutation observed associated with MELs, we have not expanded on the gene-conversion argument in the text which we feel might misdirect the reader.

5) It is important that 57% of AID and 46 % of sA3G mutations occur within the promoter region compared to only 21% of EMS mutations. However, considering that the authors found 1227 and 568 MELs in the AID and sA3G treated genomes, but only 1 for MMS, a simple analysis would reveal that the MMS mutations in the promoter region compared to the total frequency of MMS-induced MELs is extremely high. Could the authors clarify or discuss this point better, so that the reader does not get misled by a simple analysis, if that is the case?

The compact nature of the yeast genome makes the proportion of sequences associated with “promoter” function much higher than in mammalian cells, thus up to 20% of the genome corresponds to promoters. Fully random mutation will therefore result in 20% of mutations occurring on promoters, which is the frequency seen in EMS treated samples. We have made a note in the text to clarify this.

Our analysis reveals that the deaminase induced mutations are skewed towards promoter regions much more than expected (57% and 46% for AID and sA3G). Even within the small fraction of the genome corresponding to promoters, the regions with high density of mutations (that we define as MELs) are confined to even smaller regions, which when added up account for just 1.5% of the whole genome. This is a remarkable skewing that clearly deviates from random which our analyses attempt to convey.

However, as the reviewers rightly question, the single MEL identified in the EMS induced mutation set is not associated with a promoter, but with the CAN1 gene body and is the result of the selection that requires at least two mutations to inactivate the both alleles at the CAN1 locus. We have adjusted the text to emphasise this point. Therefore, there are no EMS induced MELs which associate with promoter regions.